# “C.H.A.M.P. Families”: Description and Theoretical Foundations of a Paediatric Overweight and Obesity Intervention Targeting Parents—A Single-Centre Non-Randomised Feasibility Study

**DOI:** 10.3390/ijerph15122858

**Published:** 2018-12-14

**Authors:** Kristen C. Reilly, Patricia Tucker, Jennifer D. Irwin, Andrew M. Johnson, Erin S. Pearson, Dirk E. Bock, Shauna M. Burke

**Affiliations:** 1Faculty of Health Sciences, Western University, London, ON N6A 5B9, Canada; kreill2@uwo.ca (K.C.R.); ttucker2@uwo.ca (P.T.); jenirwin@uwo.ca (J.D.I.); ajohnson@uwo.ca (A.M.J.); 2Faculty of Health and Behavioural Sciences, Lakehead University, Thunder Bay, ON P7B 5E1, Canada; erin.pearson@lakeheadu.ca; 3Lawson Health Research Institute, London Health Sciences Centre, London, ON N6C 2R5, Canada; Dirk.Bock@lhsc.on.ca

**Keywords:** childhood obesity, overweight, paediatric, parents, social cognitive theory, group dynamics, motivational interviewing, theory, knowledge translation, implementation science

## Abstract

Childhood obesity represents a significant global health challenge, and treatment interventions are needed. The purpose of this paper is to describe the components and theoretical model that was used in the development and implementation of a unique parent-focussed paediatric overweight/obesity intervention. C.H.A.M.P. Families was a single-centre, prospective intervention offered to parents of children aged between 6–14 years with a body mass index (BMI) ≥85th percentile for age and sex. The intervention included: (1) eight group-based (parent-only) education sessions over 13-weeks; (2) eight home-based activities; and (3) two group-based (family) follow-up support sessions. The first section of the manuscript contains a detailed description of each intervention component, as well as an overview of ongoing feasibility analyses. The theoretical portion details the use of evidence-based group dynamics principles and motivational interviewing techniques within the context of a broader social cognitive theory foundation. This paper provides researchers with practical examples of how theoretical constructs and evidence-based strategies can be applied in the development and implementation of parent-focussed paediatric obesity interventions. Given the need for transparent reporting of intervention designs and theoretical foundations, this paper also adds to the areas of implementation science and knowledge translation research.

## 1. Introduction

Childhood obesity is a significant and persistent public health issue, affecting approximately 124 million children worldwide [1]. Numerous types of childhood obesity treatment interventions exist [2,3], and have been implemented with varying success in school [4], primary care [3,5], and family [6,7] settings. Insofar as the latter is concerned, family-centred approaches have typically targeted parental support, familial interactions, and the home environment [8,9,10,11,12]. Epstein et al., who were among the first to design and encourage the use of family-based approaches in the treatment of childhood obesity, have noted that the primary goal is to have family members—primarily parents—take a lead role in the facilitation of behaviour modification among children [10,11]. A meta-analysis of family-based childhood obesity treatment interventions (*n* = 20 studies targeting at least one family member in the intervention to support or assist health behaviour change in the child) conducted by Berge and Everts (2011) provided support for the use of this approach. Their findings showed that family-based interventions were associated with improvements in children’s body composition (i.e., body mass index [BMI], standardised BMI [BMI-*z*], percentage overweight, and BMI percentile) as well as dietary outcomes (i.e., dietary intake and sugar-sweetened beverage intake) and physical activity. However, the authors did note that the research in this area was limited by a lack of sample diversity, high attrition rates, and inconsistencies with BMI measurements across studies [6]. Berge and Everts (2011) also identified a need for theory-driven research in the field of family-based childhood obesity intervention research.

Ample evidence supports the conclusion that parents are critical in the success of paediatric weight management programmes [9,13,14,15,16,17]. Given such evidence, interventions targeting *parents* as the primary agents of change in paediatric obesity treatment programmes have garnered increasing attention in the literature [13,17,18,19,20,21]. The work of Golan et al. [8,9,14,15] focussing on the use of parents as the exclusive intervention targets in the treatment of childhood obesity has been particularly influential. In their longitudinal study comparing a parent-only childhood overweight/obesity intervention (14 one-hour support and education group sessions over one year for parents) to a child-only intervention (30 one-hour group education sessions over one year for children), Golan and Crow (2004) found a greater reduction in percent overweight in children from the parent-only intervention than children who participated in the child-only intervention at one, two, and seven-year follow-up assessment points [9]. In a subsequent study, Golan et al. [15] evaluated the effectiveness of a six-month (16 one-hour support and education group sessions and monthly individual sessions) childhood obesity treatment intervention directed exclusively at parents versus one of the same length and structure directed at both parents and children [15]. Again, the findings demonstrated that the parent-only intervention was more effective than the similar intensity programme that included children as active participants [15]. More recently, a 2014 systematic review of randomised controlled trials (RCTs; *n* = eight studies) comparing parent-only and parent-child childhood obesity treatment interventions determined that parent-only interventions were equally or more effective than child-only interventions in reducing children’s BMI-*z* and caloric intake, improving family eating habits, and enhancing mental health outcomes among children [16]. There is also evidence demonstrating that parent-focussed childhood obesity interventions may be more cost-effective than traditional family-based (i.e., parent and child-focussed) interventions, as they are generally less expensive to implement and require fewer resources [16,22].

The evidence supporting parent-only interventions for childhood obesity is compelling. However, there is a lack of reporting in the literature related to the development and implementation of these interventions [23,24]. Without explicit descriptions of the interventions, it is difficult to identify how and why specific intervention components have been implemented, and how intervention principles and strategies can be adapted and translated into practice or future studies [16,18,25]. Theory, for example, is a critical aspect of childhood obesity intervention design, as it provides grounding for and explanation of the mechanisms through which certain variables are expected to produce behaviour change [26]. Researchers have identified a paucity in the application of theory in family-based childhood obesity interventions [23], and recommendations for theoretically grounded interventions in this area have been well documented in reviews of the literature [6,7,17,23]. A recent qualitative meta-synthesis of child and adolescent obesity family-based interventions conducted by Alulis and Grabowski (2017) showed that the authors of only 31.4% of the reviewed studies reported the use of theory in the development of the intervention [23]. In addition, the authors noted that among the studies that included a theoretical foundation, many lacked transparency in reporting with regard to how the theory was applied within the context of the intervention [23].

Findings outlined in qualitative studies provide additional support for the conclusion that greater parental involvement in childhood obesity interventions is warranted [27,28,29]. The original Children’s Health and Activity Modification Program (C.H.A.M.P.) was a four-week, family-based programme that targeted children with obesity (the day-camp ran Mondays–Fridays from 09:00–16:00) as well as their parents (parent/caregiver sessions ran on Saturdays from 10:00–14:00; see Martin et al. [30] for a full description of the programme). C.H.A.M.P. differed from other family-based paediatric obesity programmes in that it was developed on the basis of group dynamics theory and utilised evidence-based group dynamics strategies to develop cohesive (i.e., ‘true’) groups with both children and parents [30]. While the results of the pilot programme were positive with regard to child BMI-z [31], cardiovascular indices [32], child and parent-reported quality of life [31], and self-efficacy [33], some parents noted that future programmes should include more education and opportunities for parents, as well as greater parental accountability [27]. Interestingly, children who participated in C.H.A.M.P. also expressed a desire for increased support and involvement from their parents in order to initiate and maintain their lifestyle modifications [34]. On the basis of the strong empirical evidence in support of parent-focussed interventions in the treatment of childhood obesity, our research team designed and implemented an intervention entitled “C.H.A.M.P. Families”; a 13-week group and community-based educational programme targeting parents of children aged six to 14 years with overweight and obesity. Similar to C.H.A.M.P., and given the documented lack of reporting related to the use of theory in family-based childhood obesity interventions [23,24], C.H.A.M.P. Families was designed and implemented on the basis of a unique combined theoretical approach. Thus, the purpose of this paper is to provide a detailed description of this parent-focussed childhood overweight and obesity intervention, as well as the theoretical framework that was used in its design and implementation.

## 2. Materials and Methods

The following section is divided into two subsections, including: (1) a description of the C.H.A.M.P. Families intervention in accordance with the Template for Intervention Description and Replication (TIDieR) checklist and guide (Appendix A, [35]), and (2) a detailed overview of the theoretical foundation used in the design and implementation of the intervention.

### 2.1. Intervention Description

#### 2.1.1. Intervention Design

A single-centre, single-group, non-randomised, prospective feasibility study design [36,37] was used to evaluate the preliminary effectiveness and overall feasibility of C.H.A.M.P. Families, which was a group and community-based programme delivered via eight sessions over a 13-week period to parents of children six to 14 years of age with overweight or obesity in Ontario, Canada. The study was approved by the Health Sciences Research Ethics Board at a large Canadian university (Project ID# 108826) and retrospectively registered with an International Standard Registered Clinical Trial Number on April 24, 2018 (ISRCTN #10752416).

#### 2.1.2. Participant Recruitment and Eligibility

Participants were recruited over a four-month period (May–September, 2017). Multiple recruitment strategies targeting parents were utilised, including newspaper and radio advertisements, social media advertisements, posters displayed in various community settings (e.g., libraries, local businesses, family health clinics), and study pamphlets and posters delivered to community paediatricians and family physicians. Parents were eligible to participate if: (a) they had a child between the ages of six and 14 years at baseline; (b) they had a child with a BMI greater than or equal to the 85th percentile for their age and sex [38]; and (c) both the child and parent were able to speak, read, and understand English. All parents and caregivers, including those living in separate homes, were invited to attend the programme and participate in the study if interested and eligible. Families were excluded from the study if: (a) the child did not have a BMI ≥85th percentile for age and sex, (b) they did not provide written consent or assent, (c) they were unable to read, speak, or understand English, and/or (d) the child had a medical condition or used a medication that impacted physical activity levels and/or other study outcomes.

#### 2.1.3. Intervention Description

The C.H.A.M.P. Families intervention consisted of three main components: (1) eight group-based (parent-only) education sessions delivered over the course of 13 weeks, (2) eight home-based (family-directed) activities that were ‘assigned’ by the research team following each of the group-based sessions, and (3) two group-based follow-up support sessions for parents and children following completion of the formal intervention.

Group-based (parent-only) component. Parents were invited to attend a total of eight *parent-only* education sessions on Monday evenings from 18:30–20:00 from September to December 2017, which were held in a boardroom at a local YMCA facility in a large city in Ontario, Canada. The intervention was 13 weeks in total. To date, researchers have been unable to discern a clear dose-response relationship in terms of weight-related outcomes in childhood obesity interventions [39]. Further, given that family-based paediatric obesity interventions have been associated with high rates of attrition [18,24] and are time and resource-intensive, C.H.A.M.P. Families was created as a low-cost, low-intensity treatment intervention (i.e., 12 hours over 13 weeks) in an effort to increase participant retention and overall feasibility. This intervention was also designed so that the first four sessions were held weekly, and the remaining four sessions were held bi-weekly (an additional week was added to accommodate a statutory holiday). Sessions were offered less frequently as the programme progressed to avoid a reliance on the group and encourage participants to utilise their own abilities, as well as the skills discussed in class, with their families in the home environment [40,41].

Each session was group-based and included in-class discussions and presentations pertaining to several lifestyle, environmental, and social factors related to child and family health. Topics included, but were not limited to, healthy eating, physical activity, sleep, screen time, sedentary behaviour, family communication, bullying, effective parenting, and mental health (see Table 1 for a complete overview of session titles, speakers, topics, and materials/information delivered to parents). At the start of each session, parents were asked to signed an attendance sheet and were provided with materials (i.e., handouts and resources) for that session. Attendance was tracked by the Project Coordinator. Programme content was delivered verbally to the group by content experts (i.e., a researcher, health professional, and/or other expert in the area[s] of interest) and community organisations, and supplemented with slide presentations and printed resources. Printed materials—existing (e.g., Canada’s Food Guide [42]), adapted (e.g., Socioenvironmental Framework for Promoting Lifestyle Behaviours in Children [43]), or created for the purpose of the programme (e.g., C.H.A.M.P. Families Community Resources Handbook)—were provided to parents during or following each group session. Sessions were educational and interactive in nature, and included activities, evidence-based strategies, and group brainstorming related to how the information discussed in the group setting could be personally applied or used with children and family members in the home environment (intervention materials and resources available upon request). Each 90-min session followed the same general structure: (a) a 10-min review and group-based discussion related to the previous week’s goal-setting worksheet and/or readings, plus guest speaker introductions and announcements, (b) a 60-min interactive education session, (c) a 10-min question, answer, and discussion/reflection period, and (d) a 10-min overview of the following week’s session topic(s), home-based goal setting worksheet, and associated readings and resources. Parents were provided with binders to store and organise all of the programme documents and homework activities. The C.H.A.M.P. Families programme was offered at no cost to participants. Parking was free, and complimentary YMCA child-minding and drop-in activity-based programming were available for children (including siblings) during each parent session. Healthy snacks and drinks were provided at each session. Outside of the intervention, the Project Coordinator communicated with participants on a regular basis; participants received email reminders of upcoming sessions and were contacted via email and/or telephone to schedule home visits.

Home-based (family) component. At the end of each group-based session, parents were assigned a home-based goal setting worksheet to complete with their families. The worksheets were adapted from those used in an evidence-based obesity prevention programme for parents in the United States [44], and were intended to support and reinforce the concepts discussed in the group sessions. The worksheets contained several questions (closed and open-ended) regarding their family’s health behaviours and habits in the home environment. They prompted parents to consider various health behaviours, their family’s current habits and beliefs, as well as facilitators and impediments to change. The worksheets instructed parents and children to work together to set at least one “S.M.A.R.T.” (i.e., specific, measurable, attainable, relevant, and time-related [45,46,47,48]) goal related to a health behaviour and develop a family action plan, collaboratively, to reach the agreed upon goal. Participants were encouraged to share their family experiences with the group at each subsequent parent session, and submit their worksheets to researchers in order to enhance accountability and track participants’ completion of the home-based activities. The first five goal-setting worksheets pertained to the weekly educational topic(s) including: family meals; fruit and vegetable consumption; screen time and eating; physical activity; and sleep. During the last three sessions, parents were asked to revisit, with their families, one of the previously completed family goal-setting worksheets to either revise a previous goal or set a new goal. A sample of the goal setting worksheets were piloted by our team with a small group of parents (*n* = five) of children aged six to 14 who were not involved in the study.

Group-based (family) follow-up support. Two, two-hour group-based “C.H.A.M.P. Families Booster Sessions” for parents, children, and additional family members were held at three and six months post-intervention (i.e., March and June 2018). The primary purpose of the booster sessions was to provide parents and children with fun, family-focussed, active opportunities to reconnect and socialise in a group environment following the formal intervention. The first session consisted of a family cooking and food literacy class hosted by a local not-for-profit organisation, and the second was a physical activity-based obstacle course held at a local business. At each booster session, parents and/or children were provided with helpful resources (e.g., kitchen utensils and pedometers) and information about family-friendly community organisations and activities (e.g., healthy recipes, pamphlets for summer camps). Follow-up support was also offered in the form of post-intervention contacts (i.e., monthly e-mails and telephone calls) from the researchers to outline the details of the booster sessions and schedule each family’s six-month follow-up data collection visit, after which formal contact with participants ceased.

#### 2.1.4. Intervention Providers

The C.H.A.M.P. Families Research team consisted of five professors with several years of experience in childhood obesity research, one paediatrician who specialised in childhood obesity treatment, and one PhD-level graduate student who served as the Project Coordinator. One member was a Certified Professional Co-Active Coach who delivered a seven-hour motivational interviewing (MI) training session to the Principal Investigator and Project Coordinator. These team members consistently used an MI spirit (e.g., expressing empathy, drawing upon parents’ own expertise to apply content learnings in their families [49,50]) with parents, as well as specific techniques (e.g., asking open-ended questions, affirming/acknowledging participants’ experiences, reflective listening, and summarising what parents shared) throughout all aspects of the intervention. Guest speakers who were invited to deliver content to participants had appropriate experience and credentials (e.g., registered dietician, registered nurse), and in some cases, were involved in the original C.H.A.M.P. program. Individuals from community organisations (i.e., Heart and Stroke Foundation, YMCA) had been employed by these organisations for many years and had experience delivering group presentations. All guest speakers sent slide presentations to the Project Coordinator to review content in advance.

#### 2.1.5. Feasibility Assessment

The feasibility of the intervention is currently being assessed using RE-AIM [51,52,53], a framework applied in both the design and evaluation of health behaviour interventions [51,52,54]. RE-AIM consists of five dimensions: *reach* (i.e., the proportion and representativeness of individuals participating in the intervention [51,52,54]), *effectiveness* (i.e., the impact of the intervention on study outcomes [52,54]), *adoption* (i.e., the proportion and representativeness of intervention agents and settings that are willing to initiate the intervention [52,54]), *implementation* (i.e., fidelity of intervention delivery to the intervention protocol, including time and cost of intervention [52,54]), and *maintenance* (i.e., the degree to which intervention participants maintain behaviour change over time and, at the setting level, the extent to which the intervention is sustained over time [51,52,54]). Specific objectives within the ‘effectiveness’ domain of RE-AIM are to assess the impact of the intervention in relation to several child and family outcomes including: (a) children’s standardised body mass index (BMI-*z*; primary outcome); (b) parents’ BMI; (c) children’s health-related quality of life (HRQoL); (d) children’s general health and well-being; (e) children’s physical activity levels and sedentary time; (f) family cohesion, communication, and satisfaction; (g) parental self-efficacy related to engaging children in healthy eating and physical activity; and (h) children’s and parents’ overall perceptions of the programme and its impact on family health and well-being. Data pertaining to the RE-AIM dimensions were collected at various time points and on an ongoing basis prior to, during, and following the formal intervention; a publication outlining the feasibility analysis is currently in preparation. Table 2 provides an overview of the data collection for each RE-AIM dimension.

Demographic information about parent(s) and children, as well as data pertaining to the primary and secondary outcomes noted above, were collected during scheduled home visits with participants to protect their privacy and ensure the comfort of parents and children. Home visits occurred at four time points: baseline (≤ four weeks pre-intervention), mid-intervention (i.e., Week Six), post-intervention (i.e., ≤ two weeks post-intervention), and six-months post-intervention (i.e., June 2018). Each visit was 45 to 90 min in duration. Lastly, qualitative data pertaining to participants’ perceptions of and experiences in the programme were collected via focus groups with both parents and children (separately) and held at the YMCA during the last group-based session of the formal intervention. The general purpose of the focus groups was to explore parents’ and children’s perspectives of the impact of C.H.A.M.P. Families, as well as their recommendations for future interventions. All of the focus groups were approximately 75 min in duration, audio-recorded, and moderated by researchers involved in the study.

### 2.2. C.H.A.M.P. Families Integrative Theoretical Foundation

Social cognitive theory (SCT; [55,56,57,58,59]) was selected as the primary underlying theoretical model for C.H.A.M.P. Families. Broadly speaking, SCT aligns with health promotion principles, as it promotes the concept of enabling individuals to exercise control over their own health [55,56,59], and has been applied in interventions targeting a number of lifestyle behaviours including, but not limited to, diet and physical activity [60], smoking cessation [61], and alcohol consumption [62]. In relation to paediatric obesity interventions, SCT was identified by Alulis and Grabowski as the most common theoretical foundation in paediatric obesity interventions [23]. Given its emphasis on social learning [55,56,58,63] and that children learn health behaviours through observing and imitating (i.e., learning from) their parents [64,65,66], SCT is particularly suitable for interventions in which parents are targeted as family role models [8,17,43].

Within the SCT model of health behaviour, there are four determinants that can function both independently and in concert to regulate and predict behaviour: self-efficacy, outcome expectations, goals, and socio-structural factors [55,56,63,67,68]. Self-efficacy (i.e., a person’s belief and confidence in his or her ability to exercise control over their behaviours [55,56,63,67]) impacts health behaviour directly and through its influence on the other determinants [55], in that individuals with higher self-efficacy have greater expectancies of positive outcomes, set higher goals, and perceive fewer impediments to the desired behaviour [55,63,67]. Outcome expectations and goals also operate to impact behaviour directly; however, goals are influenced by perceptions of socio-structural factors and outcome expectations [55].

Self-efficacy is the focal determinant of the SCT model and an important consideration in the development of paediatric obesity interventions. Previous research has demonstrated that many parents lack confidence in their ability to manage children’s weight-related behaviours [69,70], and increasing parental self-efficacy is associated with reductions in children’s BMI [21,71]. According to Bandura [63], there are four primary sources of efficacy beliefs, all of which were targeted in the C.H.A.M.P. Families intervention: mastery experiences (i.e., “performance accomplishments” [63,68,72]), vicarious experience [63,67], verbal persuasion [63,67], and physical and emotional states (i.e., “emotional arousal” [63,68]).

The second determinant of SCT, outcome expectations, corresponds to the results that people anticipate their actions will produce if they do or do not perform a specific behaviour [55,56]; the assumption is that individuals will adopt behaviours that result in advantageous or positive outcomes, and avoid those that lead to negative outcomes [55]. Bandura asserted that outcome expectations may be physical, social, and/or self-evaluative [55,68]. The third SCT determinant is goals [55], which are the results people attempt to obtain through their actions [73]. In SCT, long-term goals provide a general guide towards the desired behaviour in the future, whereas short-term goals provide greater control over current behaviours and actions [55]. Finally, socio-structural factors represent the fourth determinant, and include those factors an individual perceives to promote (i.e., facilitators) or deter (i.e., impediments) their goals, and ultimately, the desired behaviour [55]. In order to assist in the facilitation of sustained behaviour change in their children and families, it is important that parents are able to identify the barriers obstructing certain behaviours and develop a plan to overcome them [55,74,75].

Within the general framework of and in addition to the four core SCT constructs described above, evidence-based group dynamics principles [30,76,77] and MI techniques [49,50] were also used in the design and implementation of the C.H.A.M.P. Families intervention. In general, group dynamics refer to the interactions and processes that occur within and among members of a group, as well as how individuals behave in group settings [77,78]. With regard to obesity, group dynamics strategies such as social support, collective problem solving, and group goal setting have been shown to be associated with improvements in body composition outcomes [79,80,81,82]. Furthermore, RCTs have shown that group-based interventions are more successful in decreasing children’s BMI-*z* than individual treatments [83,84].

As noted above, group dynamics theory was used as the foundation for the original C.H.A.M.P. intervention developed by members of our research team [27,30,31,32,33,34]. More recently, researchers have suggested that group-based treatments for childhood obesity targeting parents as agents of change are more effective than individualised treatments [83,84,85]. As such, the use of evidence-based group dynamics strategies in the development of an intervention targeting parents was deemed on both theoretical and empirical grounds to be an important change to the original child-focussed C.H.A.M.P. study.

Given its emphasis on self-efficacy, goal setting, and behaviour change, MI has also been utilised in SCT-based interventions [86,87,88]. MI is defined as “a person-centred counselling style for addressing the common problem of ambivalence about change” (p. 29, Miller and Rollnick [50], 2013). Thus, the primary aims of MI are to facilitate behaviour change by enabling individuals to identify their goals, examine and resolve their ambivalence to change, activate motivation for change, and ultimately make plans to change their behaviour [49,50]. Evidence from literature reviews suggests that MI is an effective strategy in addressing paediatric obesity [17,89,90,91]. Specifically, implementing MI with parents of children with overweight and obesity has been found to be effective in terms of reducing BMI in both feasibility [92] and RCT [93,94] studies. While MI was created and intended for in-person, individual counselling, there is also evidence to suggest that MI strategies and techniques can be delivered effectively via text-based materials and resources [44,95,96], as well as in group-based settings [97,98]. Given this rationale, MI techniques were strategically utilised within C.H.A.M.P. Families group sessions and in the home-based worksheets, as outlined below.

### 2.3. Applications of Specific Theoretical Constructs and Strategies within the C.H.A.M.P. Families Intervention

The following section contains an overview of the application of the four SCT constructs described above, as well as a description of the adaptation of specific group dynamics strategies and MI techniques (emphasised using italics) that were used in the development and implementation of the three C.H.A.M.P. Families components. Table 3 provides definitions of each theoretical component that was applied in the design and implementation of C.H.A.M.P. Families.

#### 2.3.1. SCT Determinant #1: Self-Efficacy

Group-based (parent-only) sessions. Enhancing parents’ confidence in their ability to support and assist in their family’s health behaviour modifications is crucial to create action towards change [55,56], and was applied in C.H.A.M.P. Families by targeting the four main sources of efficacy [63,67,68]. Specifically, the research team members used an MI spirit [50] to enhance participants’ self-efficacy through verbal persuasion [63] by offering suggestions when requested, verbal encouragement, and/or reinforcement that the parents had the personal resources, tools, and knowledge necessary to help their child(ren) achieve the desired health behaviours [55]. Similarly, guest speakers and many of the participants themselves were also sources of verbal persuasion to other participants. Another method of enhancing parental self-efficacy in C.H.A.M.P. Families was the deliberate creation of potential mastery experiences [63], whereby participants were provided with opportunities to practice, apply, and experience success related to new information and health behaviours. For example, after participating in the nutrition-related sessions, parents were invited to attend an optional, guided, hands-on grocery store tour led by a registered dietician. During the tour, parents were encouraged to practice reading nutrition labels and ingredient lists, identify healthy and ‘budget-conscious’ foods, compare and identify different and potentially novel/unique healthy food items, and ask questions and engage in discussions with the dietician and other participants. All of these exercises allowed participants to apply the knowledge gained during group sessions in a “real life” setting. From a group dynamics perspective, this activity is also reflective of how strategies such as *guidance, observational learning,* and *interpersonal learning* were applied and utilised between parents as well as between researchers/expert and parents [30,77].

Finally, during the first group session, in which an MI outcomes-based activity took place (described in more detail below), many parents acknowledged that modifying family health behaviours and having weight-related discussions with children were intimidating and anxiety-provoking (i.e., they had the potential for eliciting negative emotions [63,72]); arguably, these and other concerns identified by parents were likely to hinder parents’ efficacy to serve as agents of change for their families. Thus, the C.H.A.M.P. Families researchers worked to ensure that the issues and needs identified by parents were addressed throughout the programme in an effort to enhance parental self-efficacy. For example, several tips, strategies, and resources for positive communication with children about health, nutrition, body image, body size, and weight were provided to parents in an effort to address parents’ concerns and support positive family cohesion and dynamics [100,101,102]. From an MI perspective, this is also an example of how the researchers applied *affirmation* (i.e., acknowledgement and validation [49]).

Home-based (family) activities. The home-based worksheets were specifically designed and adapted [44] to support participants’ self-efficacy using MI techniques including open-ended questions [49], which were designed to guide parents through identifying their views on the importance of and personal values associated with a specific health behaviour. For example, the worksheets related to children’s sleep contained the following questions: “*Think about your child’s bedtime…On a scale of 1 to 10, how important is it to you that your child gets enough sleep each night?*”; “*What is important to you about your child getting enough sleep each night?*”; and “*What does your child’s bedtime routine look like (please include approximate durations)?*” [44]. Additional questions encouraged parents to identify current behaviours and factors that both prevented and promoted the behaviours that they deemed to be important.

Group-based (family) follow-up support. A primary goal of the booster sessions was to provide additional opportunities to enhance parents’ *and* children’s self-efficacy for engaging in health behaviours after completion of the formal intervention. The sources of self-efficacy that were targeted included vicarious experiences (e.g., participants observed the obstacle course leader and one another to learn how to complete each obstacle), mastery experiences (e.g., professional chefs demonstrated each cooking skill and then gave parents and children the remainder of the session to engage in the skills and then prepare and cook an entire meal for the group), physiological feedback (e.g., families were provided with pedometers to take home and wear during the obstacle course activity so they could track and receive instant feedback about their step counts), and verbal persuasion (e.g., participants, researchers, and instructors encouraged each other regularly and throughout both booster sessions). Booster sessions served as further opportunities to promote group dynamics strategies including *ongoing communication and interaction, guidance*, *observational learning,* and *interpersonal learning* [77].

#### 2.3.2. SCT Determinant #2: Outcome Expectations

Group-based (parent-only) sessions. As noted above, in the first group session, a member of the research team who was a Certified Professional Co-Active Coach (CPCC) and experienced MI facilitator encouraged parents to consider and share the different types of outcomes (i.e., physical, social, and self-evaluative [55]) that they expected for themselves and their children during and upon completion of the programme. One of the benefits of the co-active coaching approach is that it sets MI tenets into action [103], which was important for this activity, given that identifying outcome expectations can increase the likelihood of sustained behaviour change [55,104,105]. Specifically, the CPCC used foundational MI techniques, including *asking open-ended questions*, *affirmations*, and *reflective listening* to prompt participants to share their thoughts and expectations with the group [50,103]. The CPCC also *summarised* the information provided by participants and created a list of parent-identified outcome expectancies, which were shared in real time via a screen and projector. The participants then reviewed, discussed, and refined the list of desired outcomes. This activity was done collectively to enhance *cohesion* and promote *self-disclosure* to the group [30,77]. Once the list of expected outcomes was complete, it was circulated to parents as well as all programme staff and speakers, and was revisited by the C.H.A.M.P. Families research team frequently to ensure that all of the programme sessions were tailored to and addressed parents’ expectations.

#### 2.3.3. SCT Construct #3: Family Goal Setting

Group-based (parent-only) sessions. In addition to the exercise outlined above in which programme expectations/goals were identified, parents participated in a structured group-based goal-setting workshop that provided background information and evidence pertaining to the importance of setting group goals, as well as instruction on how to set S.M.A.R.T. family goals and action plans [45,46,47]. This session, led by the Principal Investigator, included an interactive presentation that emphasised the importance of involving the entire family in the goal-setting process to increase accountability and provide motivation for behaviour change [55]. Participants were encouraged to find a “buddy” with whom they could share their contact information to promote accountability (i.e., staying true to a promise of action [99]), which is a co-active coaching and MI tool, for family goals. From a group dynamics perspective, this also created additional opportunities for fostering *cohesion* and *social support* among parents [77]. Other group dynamics techniques that were utilised in relation to group goal setting were *information sharing* and *self-disclosure*; at the start of each session, participants discussed their experiences—including successes, challenges, and revisions to goals—with the previous week’s family goal-setting exercise [77].

Home-based (family) activities. In addition to setting a “S.M.A.R.T.” goal each week (described above), families were asked to create an action plan to achieve their goals (i.e., three specific steps that they would take to reach their goals, by when, how they would track their progress, resources or supports they would need for success, and the main reasons that the goal was important for their family).

Group-based (family) follow-up support. Informal group discussions were held with parents and children regarding progress towards the family goals set during the programme, as well as post-programme goals and action plans. However, some parents noted that it was challenging to maintain the behaviour changes following the structured group-based sessions. To address this, researchers used affirmation [50] to validate parents’ feelings. Additional group dynamics and SCT-related strategies were also applied (i.e., support, guidance, information sharing, verbal persuasion [30,63,77]) to bolster self-efficacy and encourage parents to persist.

#### 2.3.4. SCT Construct #4: Socio-Structural Factors (Facilitators and Impediments).

Group-based (parent-only) sessions. At numerous times throughout the intervention, parents were encouraged by the expert speakers and researchers to identify and discuss barriers for health behaviour changes among their children, as well as plans to overcome such obstacles. This resulted in opportunities for *collective problem solving* [30,40] and *self-disclosure* (i.e., the revealing of personal information to a group), which are evidence-based group dynamics strategies that are used to foster *social support* and *cohesion* [77]. Additional group dynamics constructs used—particularly in cases wherein parents had experienced similar impediments and identified strategies to overcome them—were *information sharing*, *guidance*, and *interpersonal learning* [77]. For example, one parent described numerous situations in which her child refused to eat vegetables. To help overcome this impediment, other parents offered several potential solutions that had been effective in their families, including using a spiraliser device to make vegetable noodles, incorporating the child into food and meal preparation, growing vegetables in a garden, and grating vegetables into sauces. In a session related to mental health, another parent shared personal information about her child who was being bullied at school, as well as insights, experiences, and resource information related to the process of reporting bullying and advocating for children within the local school system. In addition to discussing challenges, participants who had identified factors that led to a positive impact on their family’s health behaviours (i.e., facilitators) were asked to share their experiences in the same way.

When facilitating group discussions, researchers regularly employed MI techniques to empower and motivate participants. For example, when discussing the barriers to behaviour change, the researchers used *open-ended questions* such as, “*What is hard about that?*” and “*What would be the first step?*”, to help guide the conversation towards a solution. The discussions about impediments also created an opportunity for participants to receive *affirmation* from both the researchers and other parents [49].

Home-based (family) activities. The home-based worksheets included questions that encouraged parents to reflect upon the facilitators and impediments to the lifestyle behaviours discussed in the group sessions. For example, in terms of facilitators for family meals, parents were asked open-ended questions such as: “*Think back to the last time you had a family meal that went really well. What was good about it? What needs to happen to have meals that go well more regularly?*” [44]. Examples of MI *open-ended questions* about impediments to adequate sleep for children and to encourage ‘change talk’ [50] were, “*What types of things typically prevent your child from getting enough sleep each night? What ideas do you have for overcoming these barriers?*” [44]. These questions and others were intended to empower and prepare participants for change by having them explore their own ideas and experiences [49,50].

#### 2.3.5. Additional Group Dynamics and Motivational Interviewing Strategies Applied within the C.H.A.M.P. Families Intervention

Group-based (parent-only) sessions. In addition to, and overlapping with some of the theoretical constructs discussed above, additional group dynamics strategies were targeted regularly throughout the programme that warrant additional emphasis. These include: (1) *distinctiveness*; (2) *proximity*; (3) *ongoing interactions and group-based activities;* and (4) *ongoing communication, feedback, and social support* [30,76]. Group dynamics research suggests that groups with a greater sense of *distinctiveness* tend to be more cohesive than groups that do not perceive themselves to be distinct from or unique in comparison to other groups [30,76]. C.H.A.M.P. Families was a new programme with a unique philosophy that focussed on creating an empathetic, safe, and judgement-free environment [50] wherein parents were supported in sharing information about their personal experiences and challenges. Given its specific focus and unique group-based environment, the C.H.A.M.P. Families group was likely different from any other group(s) to which parents belonged (e.g., family, work, social), making it inherently *distinct*. Furthermore, the group environment and classroom layout (i.e., desks arranged in a U-shape to encourage face-to-face communication, with participants often sitting in the same seats throughout the programme) ensured that participants were in close and consistent physical *proximity* to one another other, which is a group dynamics factor related to *group cohesion* and *social support* [30,106]. Attending educational sessions enabled participants to meet and interact with people who shared similar circumstances, which is something that parents in previous paediatric obesity interventions have deemed important and valuable [34]. Beyond several group-based interactions and activities, these sessions were the main mechanism through which participants could provide and receive encouragement, guidance, and support from their peers [30,76].

While parents were the primary intervention targets of C.H.A.M.P. Families, children were invited to attend free, group-based programming offered at the YMCA (e.g., swimming, sports, board games, etc.) during the parent-only sessions. While the theoretical constructs described herein were not used explicitly with children, many parents reported that their children had very positive experiences in the group-based programming, including developing new friendships and engaging in new activities with other similar peers. Several children from C.H.A.M.P. Families attended the optional programming regularly, thus creating the same opportunities for: (1) *ongoing interactions and group-based activities*; (2) *ongoing communication*, *feedback*, *and social support*; and (3) *cohesion* [30,76]. This is an important programmatic feature that warrants mention, given the positive impact on the children, and also on the parents who were leaving their children to participate in the weekly group-based parent sessions.

Home-based (family) activities. In addition to the home-based activities (i.e., discussions, goal setting, etc.) led by parents, another aspect of C.H.A.M.P. Families that took place at home was the four data collection visits with the Project Coordinator. From a group dynamics perspective, these visits were important opportunities to continue the *ongoing communication, feedback, and social support* for families throughout and following the intervention [30,76]. Further, given that the Project Coordinator was one of the research team members trained in MI, she approached each visit and conversation as a partnership with an MI spirit (i.e., creating a participant-centred partnership, with the attitude of acceptance, compassion, and evocation; Miller and Rollnick, 2013) and actively applied MI techniques (i.e., OARS: *open-ended questions*, *affirming, reflective listening*, and *summarising;* Miller and Rollnick, 1991, 2013). Participants were also provided with the contact information of the Programme Coordinator, Principal Investigator, and guest speakers, and were informed that they could contact these individuals at any time if they had questions and/or required additional support.

Group-based (family) follow-up support. The activities selected for the booster sessions (i.e., cooking class and obstacle course) were purposefully interactive and group-based, often requiring participants to not only work together within their family unit, but also with other families and members of the research team. In addition to serving as opportunities for *interpersonal* and *observational learning* [77], these experiences provided parents and children with time to reconnect, facilitating *cohesion* and *ongoing communication, feedback, and social support* [30,76]. Again, given the MI training of the research team, an MI spirit formed the foundation for all of the conversations and group discussions with participants [50].

## 4. Discussion

The treatment of childhood overweight and obesity requires complex behavioural and theory-based interventions [13,17,43]. This article provides a rationale for and detailed description of the C.H.A.M.P. Families intervention, as well as practical applications of its foundational theoretical framework. Previous research supports the utilisation of SCT [17,71], MI [89,90,92,107], and group dynamics theory [30,85,108] in childhood obesity interventions, and this paper addresses an important gap in the literature by providing a detailed description of the process by which theoretical constructs are adapted, used, and integrated in a unique intervention targeting parents as the primary agents of change [3,7,23,24]. The transparent reporting of interventions and their theoretical underpinnings is critical not only for study replication purposes, but also to highlight the factors that might have an influence on the intervention’s success or failure [25,26].

Despite our extensive recruitment efforts, which spanned four months, we were only able to recruit 11 families to participate in C.H.A.M.P. Families (six dyads and five triads; *n* = 11 children and 16 parents/guardians). Some examples of ‘lessons learned’ from a recruitment and retention perspective include extending the recruitment period, directing additional efforts towards the messaging and ‘marketing’ of the programme, and including and highlighting greater child involvement and additional family-based hands-on activities. As noted above, the next step is to evaluate the feasibility (including the effectiveness) of the intervention using RE-AIM [51,52,53,54]. Numerous guidelines, templates, and checklists have been developed to improve reporting, and subsequently, replicate interventions [35,109,110,111]. TIDieR [35] and RE-AIM [51,52,53,54] represent tools that can serve to improve the speed, quality, and impact of public health interventions with the goal of translating research into practice [35,112]. Providing a thorough description of a theory-guided intervention such as C.H.A.M.P. Families via TIDieR, and using RE-AIM to evaluate its feasibility, may contribute to improvements in intervention content, delivery, intensity, and dose, as well as participant recruitment, retention, and adherence [53]. Through the dissemination of our feasibility analysis using RE-AIM, specific issues and limitations—as well as suggestions and potential improvements for future researchers and practitioners—regarding programme implementation and participant recruitment will also be addressed. If sufficient evidence of intervention feasibility is found, future steps may also include using a framework such as the PRACtical planning for Implementation and Scale-up guide (PRACTIS guide [113]) to continue our discussions with community stakeholders regarding the potential to scale up C.H.A.M.P. Families (or an adapted version including components from the original C.H.A.M.P. intervention [27,30,31,34] that takes into account the needs/vision of the stakeholders involved) in the community and beyond.

## 5. Conclusions

This article provides researchers and health professionals with practical information and guidance related to the process of developing a theory and evidence-based childhood overweight and obesity treatment intervention targeting parents as the primary agents of change. Reporting detailed intervention descriptions, as well as information related to the adaptation and use of theoretical constructs, is not only useful from an implementation science perspective [113], but also addresses important gaps in the literature. 

## Figures and Tables

**Table 1 ijerph-15-02858-t001:** Children’s Health and Activity Modification Program (C.H.A.M.P.) Families intervention components, providers, and topics/activities.

Timeline (MM/YY)	Intervention Component	Intervention Provider	Topics and/or Activities
08/17	Home Visit 1	- Researcher/Project Coordinator (Masters in Public Health [MPH]; Western University)	Letters of information, consent forms, demographic form, data collection
09/18	Education Session 1 “Welcome to C.H.A.M.P. Families”	- Principal Investigator (PhD; Western University) - Certified Professional Co-Active Coach (CPCC) & Motivational Interviewing Facilitator (PhD; Western University) - Hospital-based Paediatrician (MD; London Health Sciences Centre) - Public Engagement Coordinator (PhD; Obesity Canada)	Topics: Childhood growth and development, childhood overweight and obesity, weight bias, stigma, communication, goal setting, programme expectations
09/18	Education Session 2 “Setting The Table For Healthy Eating At Home”	- Principal Investigator (PhD, Western University) - Registered Dietician (RD) (PhD; Brescia University College)	Family goal setting, time management, healthy eating, family meals, parental role modeling
10/18	Education Session 3 “Nutrition by The Numbers”	- Registered Dietician (RD) (PhD; Public Health Unit)	Diet, nutrition, serving sizes and portion control, meal planning, healthy grocery shopping on a budget, sugar-sweetened beverages, goal setting, reading food labels, grocery store tour (optional)
10/17	Education Session 4 “Get Up and Get Moving”	- Researcher/Project Coordinator (MPH; Western University) - Exercise Instructor/Programme Director (PhD; Goodlife Kids Foundation)	Screen time, sedentary behaviours, physical activity, family-friendly exercise (circuit) demonstration and group activity
10/17	Home Visit 2	- Researcher/Project Coordinator (MPH; Western University)	Data collection
10/17	Education Session 5 “Family Communication, Mental Health, and Sleep: Let’s Talk About It”	- Public Health Nurse (Registered Nurse [RN]; Public Health Unit)	Family cohesion, sleep and sleep hygiene, bullying communication, resilience and mental health;
11/17	Education Session 6 “Cooking with Kids”	- Professional Chef (Community Food Education Centre—Local Organisation)	Food skills, meal preparation, food safety, age-appropriate activities for kids, nutrition, nutrition literacy
11/17	Education Session 7 “Community Connections”	- Area Administrator (Heart and Stroke Foundation) - Membership Representative (YMCA)	Media literacy, marketing of unhealthy foods and beverages to kids, health advocacy, community resources, creating an awareness campaign (group activity)
12/17	Education Session 8 “The Grand Finale: Family Celebration and Certificates”	- Principal Investigator (PhD; Western University) - Researcher/Project Coordinator (MPH; Western University)	Recap/summary of program, family celebration, group discussions, family award presentations, focus groups, farewell
12/17	Home Visit 3	- Researcher/Project Coordinator (MPH; Western University)	Data collection
03/18	Booster Session 1	- Professional Chefs (x3) (Community Food Education Centre—Local Organisation) - Principal Investigator (PhD; Western University) - Researcher/Project Coordinator (MPH; Western University)	Food skills, nutrition literacy, healthy eating, food safety, food preparation, family cooking class
06/18	Booster Session 2	- Group Exercise Instructors (x2) (Fitness and Recreational Centre) - Principal Investigator (PhD; Western University) - Researcher/Project Coordinator (MPH; Western University)	Physical activity, active play, family obstacle course
06/18	Home Visit 4	- Researcher/Project Coordinator (MPH; Western University)	Data collection

**Table 2 ijerph-15-02858-t002:** Description and timeline of data collection prior to, during, and following the 13-week C.H.A.M.P. Families intervention. BMI: body mass index, BMI-*z*: standardised BMI, RE-AIM: Reach, Effectiveness, Adoption, Implementation, Maintenance.

Outcome(s)	Measure(s)	Data Collection Time Points				
		Baseline (≤Four Weeks Pre-Intervention)	Mid-Intervention (Week Six)	Final Session of Intervention (Week 13)	Post-Intervention (≤Two Weeks after Final Session)	Follow-Up (Six Months Post-Intervention)
Demographic Variables	*Parent*: Age, sex, ethnicity, marital status, education level, household income, relationship to child *Child*: Age, sex, estimated age that weight became an issue, years child has been overweight	X X				
Feasibility Outcomes, RE-AIM Dimensions [51]:						
Reach	Family (parent and child) demographics compared to census demographics in London and Ontario, Canada; records of participant and/or non-participant inquiries to determine participate rate, reasons for participating, reasons for declining to participate, and the most effective recruitment methods	X				
Effectiveness	Short-term (i.e., baseline to mid and/or post-intervention) measurements of primary (i.e., child BMI-*z*) and secondary outcomes (i.e., parent BMI, health-related quality of life; general health and well-being; family communication, cohesion, and satisfaction; parental self-efficacy; and child physical activity) short-term attrition, reasons for drop out, qualitative data pertaining to programme impact (focus groups)	X	X	X	X	
Adoption (Setting and Staff)	Roles, credentials, demographic, and/or representativeness information of delivery settings and intervention agents/staff, where applicable	X	X	X	X	X
Implementation	Fidelity and adaptations to study protocol, intervention adaptations, completion of participant worksheets, and associated costs (including in-kind) of programme	X	X	X	X	X
Maintenance (Individual and Setting)	*Individual*: Six-month follow-up on primary (i.e., child BMI-*z*) and secondary outcomes (i.e., parent BMI, health-related quality of life; general health and well-being; family communication, cohesion, and satisfaction; parental self-efficacy; and child physical activity), long term (i.e., six-month) attrition *Setting*: Mixed-methods questionnaire delivered to staff and organisations (>six months post-intervention) to assess perceptions of programme and interest in future involvement					X

**Note.** ‘X’ indicates the timepoint(s) that each outcome was measured.

**Table 3 ijerph-15-02858-t003:** The theoretical constructs and strategies used in the design and implementation of the C.H.A.M.P. Families intervention.

Theoretical Component	Description
Social Cognitive Theory	
Self-efficacy	An individual’s confidence that change or attainment of a goal is achievable [55,56]
Outcome expectancy	The outcome(s) an individual anticipates their actions will produce [55,56]
Goal setting	Identification of a future outcome that is desired and establishing a plan to achieve that result [55,56]
Socio-structural factors	Factors that an individual perceives to promote or obstruct the desired behaviour [55,56]
Group Dynamics Strategies	
Information sharing	Providing information through group discussions and activities [30]
Observational learning	Developing skills through observation and imitation of others [77]
Interpersonal learning	Developing skills by interacting with other group members [77]
Guidance	Offering and accepting direction to and from the group [77]
Group cohesion	Belonging to a group and building strong relationships with group members [77]
Self-disclosure	Revealing personal information to the group [77]
Collective problem solving	Identifying problems and then developing solutions and strategies to overcome them as a group [30]
Proximity	Being within close physical proximity to the group [30]
Distinctiveness	Perceiving that the group is unique from other groups [30,76]
Ongoing communication, feedback, and social support	Sustained contact and supportive relationships with group members [30,76]
Ongoing group-based activities and interaction	Sustained participation and completion of group tasks and actions [30,76]
Motivational Interviewing (MI) Strategies	
MI spirit	Creating a participant-centred partnership, with the attitude of acceptance, compassion, and evocation [50]
Accountability	Acknowledging an individual’s intentions and promise of action [99]
Asking open-ended questions	Inviting an individual to reflect and elaborate in effort to gather information, evoke motivation, and plan a course towards change [50]
Reflective listening	Restating what an individual has said to demonstrate and/or clarify understanding its meaning and allow the individual to hear his/her thoughts or feelings again [50]
Affirming	Acknowledging the difficulty that an individual has experienced and recognising personal strengths and capacity for growth and change [50]
Summarising	Collection of reflective statements that drawn together suggest links between what an individual has said during a session and/or discussed prior [50]
Change talk	Promoting behaviour change by having individual verbalise arguments in favour of change [50]

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
