# Peer review of "“C.H.A.M.P. Families”: Description and Theoretical Foundations of a Paediatric Overweight and Obesity Intervention Targeting Parents—A Single-Centre Non-Randomised Feasibility Study"

_ijerph, 2018, doi:10.3390/ijerph15122858_

Round 1
Reviewer 1 Report
A solid paper which summarises much of the evidence for community-based paediatric obesity programs. In particular the second section of the manuscript on the theoretical foundation of CHAMP is very strong. However, details of the study in section 1 are lean and need to be complemented with further rationale and information, including group size, timelines, protocols for handling withdrawals. the abstract states that this study is unique but it is unclear what is the unique element or elements of CHAMP families. Direct comparison of CHAMP to other family-based child obesity programs which are similar and have been scaled-up is not included. Specific suggestions to improve the manuscript are also listed below.
I commend the authors on a rigorous and thorough study and a mostly very well written protocol. I would recommend this paper for publication.
Introduction
- too long, unclear how CHAMP (old or new) is unique
Methods
- CONSORT with extension to randomised pilot and feasibility trials was used as framework to report protocol but this is not a randomised study (line 147) – have the authors considered an alternative framework to be more appropriate? E.g. SPIRIT. Alternatively, please provide a rationale for the use of framework for randomised trials with a non-randomised study. Also suggest for a completed checklist to accompany the manuscript for reviewing purposes.
- trial registration, need to state whether prospectively or retrospectively registered (i.e. before or after enrolment of first participant) and state the date of registration
- ethics approval, is there a study or reference number for this?
- section 2.1.4 and 2.1.5 – unclear whether the sample size of 30 corresponds to one parent group of if there was a desired parent group size – e.g. 3x groups of 10; 2x groups of 15 or 1x group of 30. Please add these details where appropriate
- line 219, you have not previously defined MI?
- table 1, it is unclear whether sessions were talking, education based or if there were practical elements, e.g. use of food models, label reading, cooking demonstrations (session 6) – please add in more detail if possible. Please add in also a timeline, house visits and their aims, booster session information etc so that this table contains all information about the CHAMP families intervention
- please include in this section the rationale for the length of the program
- inappropriate to cite a manuscript in preparation
- table 2 is great, however why is effectiveness only measured short-term when you could measure at 6-months post intervention?
- line ~300, information about focus groups is insufficient, suggest to include more information here
- there is no detail on data analysis for the outcomes listed in table 2 and this needs to be provided
- line 401 – was achieved in CHAMP – how do you know? Should this be was applied?
- line 413 – was the grocery store tour part of the intervention or was it optional? Please add to table 1.
- line 551 – unclear why open-ended questions is italicised, as well as other italics emphasis in this section (excluding quotes, or questions)
- line 570 – please cite other programs in the community which CHAMP is different to
Discussion
- line 644 suggest to replace via with through
Conclusion
- seems to relate to section 2 of the protocol paper only, suggest to amend.
References
- please amend references with long and institutional URLs – e.g. 125, 122, 71. Instead use DOI URLs or even cache if appropriate
Author Response
Response to Reviewer 1 Comments
General Comments:
A solid paper which summarises much of the evidence for community-based paediatric obesity programs. In particular, the second section of the manuscript on the theoretical foundation of CHAMP is very strong. However, details of the study in section 1 are lean and need to be complemented with further rationale and information, including group size, timelines, protocols for handling withdrawals. the abstract states that this study is unique but it is unclear what is the unique element or elements of CHAMP families. Direct comparison of CHAMP to other family-based child obesity programs which are similar and have been scaled-up is not included. Specific suggestions to improve the manuscript are also listed below.
I commend the authors on a rigorous and thorough study and a mostly very well written protocol. I would recommend this paper for publication.
Response: We would like to thank Reviewer 1 for her/his insightful comments and suggestions. We have added details pertaining to the study rationale/uniqueness, as well as information pertaining to group size and timelines in Section 1 (please see responses to specific comments below for page numbers).
Introduction
Point 1: Too long, unclear how CHAMP (old or new) is unique
Response 1: Thank you for this comment. While we were able to remove quite a bit of original content, given the requested revisions and suggestions by both reviewers, we also added a substantial amount of new information (detailed throughout the response letters). Therefore, the manuscript is currently the 26 pages in length.
We also appreciate the opportunity to elaborate on the uniqueness of C.H.A.M.P. and C.H.A.M.P. Families, which we have now done in the Introduction (Lines 140-143 and 154-160). As noted in our manuscript, there is a lack of literature related to the use or application of theory in family-based childhood obesity interventions. When C.H.A.M.P. was originally designed, it was the first research-based childhood obesity intervention in Canada that had systematically used group dynamics theory and evidence-based group dynamics principles to effect changes in children’s health behaviours (Martin et al., 2009). Subsequently, and similar to the original C.H.A.M.P. program, C.H.A.M.P. Families was designed and implemented using a strong combined theoretical (group-based) approach (including the evidence-based group dynamics strategies noted above and outlined in the present manuscript) that is unique to this area of research. The use of such a theoretical framework as a foundation, and throughout the duration of the program, addresses a gap in the literature and serves to distinguish C.H.A.M.P. Families from other family-based childhood obesity programs.
Methods
Point 2: CONSORT with extension to randomised pilot and feasibility trials was used as framework to report protocol but this is not a randomised study (line 147) – have the authors considered an alternative framework to be more appropriate? E.g. SPIRIT. Alternatively, please provide a rationale for the use of framework for randomised trials with a non-randomised study. Also suggest for a completed checklist to accompany the manuscript for reviewing purposes.
Response 2: Excellent point. Upon further consideration and review of alternative frameworks, we have now decided to follow the Template for Intervention Description and Replication (TIDieR) checklist and guide (Hoffman et al., 2014). Specifically, given that our intent was not necessarily to provide an overview of the study protocol per se, but rather, to provide a detailed description of the intervention to enable other researchers to either replicate or build upon C.H.A.M.P. Families, the TIDieR checklist was deemed to be most appropriate for use in this study. As per the TIDieR guide specific information pertaining to the intervention providers (Lines 448-466), location (line 251), dose (Line 252-259, 428), and adherence (Lines 271-273, Lines 304-307) was added. Information that related to the study protocol and was not necessary to describe the intervention was removed (i.e., participant recruitment [Page 4], sample size [Page 4], Primary and Secondary Outcomes in Table 2 [Pages 9-10])
The application of the TIDieR checklist also compliments the secondary purpose of the manuscript, which is to describe the theoretical framework underpinning the intervention, as well as provide practical examples of how theoretical constructs and evidence-based strategies can be applied in future parent-focused paediatric obesity interventions. Lastly, please note that we have added a completed TIDieR checklist as a Supplementary file as per the reviewer’s request.
Point 3: Trial registration, need to state whether prospectively or retrospectively registered (i.e. before or after enrolment of first participant) and state the date of registration
Response 3: The trial was retrospectively registered with ISRCTN on April 24, 2018. This information was added on Lines 190-191.
Point 4: Ethics approval, is there a study or reference number for this?
Response 4: Yes, the Project ID# has been inserted on Lines 189-190.
Point 5: Section 2.1.4 and 2.1.5 – unclear whether the sample size of 30 corresponds to one parent group of if there was a desired parent group size – e.g. 3x groups of 10; 2x groups of 15 or 1x group of 30. Please add these details where appropriate
Response 4: Thank you for this comment. As noted in Response 2, given that we are no longer reporting a detailed study protocol in this manuscript (rather, a description of the intervention), we have removed the section titled “Sample size”. To clarify, however, the sample size of 30 corresponded to one parent group.
Point 6: Line 219, you have not previously defined MI?
Response: Thank you. We have now revised the manuscript to include “motivational interviewing” prior to first use of abbreviation (see Line 453).
Point 7: Table 1, it is unclear whether sessions were talking, education based or if there were practical elements, e.g. use of food models, label reading, cooking demonstrations (session 6) – please add in more detail if possible. Please add in also a timeline, house visits and their aims, booster session information etc so that this table contains all information about the CHAMP Families intervention
Response 7: Thank you for this suggestion. As per the reviewer’s suggestion, we have made several revisions to Table 1 so that it is more inclusive of all intervention components. First, an approximate timeline was added using months and year (i.e., MM/YY) to reflect the chronological order of the intervention. Second, information about the Home Visits (n = 4) and Booster Sessions (n = 2) was added. Finally, additional information regarding the practical (‘hands-on’) activities that took place during the intervention was also added. The general structure of each session, which were education-based and included verbal presentations and group discussions, is described on page 5; we have also edited this section to provide additional information and clarification (see Lines 265-307).
Point 8: Please include in this section the rationale for the length of the program
Response 8: We have now bolstered the rationale for the length of the program as per the reviewer’s suggestion (see Page 5, Lines 252-259). Generally speaking, medium-high intensity (i.e., 26-75 hour) family-based childhood obesity interventions administered over long study periods have been associated with high rates of attrition (Jang, Chao, & Whittemore, 2015; Loveman et al., 2015) and are time- and resource-intensive. C.H.A.M.P. Families was designed intentionally as a low-cost, low-intensity treatment intervention for overweight/obesity that was administered over a 13-week period (12 hours total) with goals of increasing participant retention and overall feasibility.
Point 9: Inappropriate to cite a manuscript in preparation
Response 9: Thank you. We have now removed the citation from the text (Lines 500-501) as well as the reference from the Reference list.
Point 10: Table 2 is great, however why is effectiveness only measured short-term when you could measure at 6-months post intervention?
Response 10: Thank you for this question. Effectiveness is in fact measured 6-months post-intervention as well; however, according to the RE-AIM Framework, whereas short-term impact (i.e., pre/post) of the intervention is measured under the “Effectiveness” dimension, long-term impact (i.e., 6-month follow up) of the intervention is assessed within the individual “Maintenance” dimension (please see Maintenance category, Table 2).
Point 11: Line ~300, information about focus groups is insufficient, suggest to include more information here
Response 11: Thank you for this comment. We have now added some additional information about the focus groups (Page 10, Lines 518-522). We hope this is sufficient. Given the broad scope of this mixed-methods project, additional detail pertaining to focus group methodology and analysis will be presented in forthcoming manuscripts outlining the qualitative results. This also allows the present manuscript to be more succinct and focused on describing the intervention and theoretical framework.
Point 12: There is no detail on data analysis for the outcomes listed in table 2 and this needs to be provided
Response 12: Thank you for this comment. As stated above, the purpose of this manuscript has now been revised to focus on a detailed description of the intervention and its theoretical framework for the purposes of replication and/or adaptation in future research (using the TIDieR guide; Hoffman et al., 2014) rather than a detailed protocol including data analysis for all primary and secondary outcomes. As such, we have decided only to present information pertaining to the feasibility assessment via RE-AIM in Table 2, as it is a requirement of the TIDieR checklist (and not the planned analysis for the primary and secondary outcomes; see revised Table 2 on Pages 9-10 where the timeline and measures for the outcomes have been removed, and all outcomes have been listed with the RE-AIM dimensions of Effectiveness and Maintenance). The specific methods and data analysis for primary and secondary outcomes will be detailed in subsequent manuscripts that report on intervention effectiveness for those relevant outcomes.
Point 13: Line 401 – was achieved in CHAMP – how do you know? Should this be was applied?
Response 13: Good point. We have revised “achieved” to “applied” as per the reviewer’s suggestion (see Page 13, Line 623).
Point 14: Line 413 – was the grocery store tour part of the intervention or was it optional? Please add to table 1.
Response 14: Thank you. The grocery store tour was optional (Line 635), but all other “practical elements” took place within the education sessions delivered at the YMCA. We have added the tour to Table 1 (including the fact that it was optional) as per the reviewer’s suggestion.
Point 15: Line 551 – unclear why open-ended questions is italicised, as well as other italics emphasis in this section (excluding quotes, or questions)
Response 15: The specific theoretical components (i.e., group dynamics strategies and MI techniques) that were used throughout the intervention, and outlined in Table 3, were emphasized intentionally using italics in this section of the paper. To reduce confusion, we have now inserted a brief statement on Page 12 (Line 597) noting this emphasis. We are also happy to eliminate the use of italics in this section if the reviewer and Editor feel that is more appropriate.
Point 16: line 570 – please cite other programs in the community which CHAMP is different to
Response 16: We have now edited this sentence to enhance clarity (see Page 17, Lines 794-797). The purpose of this sentence was not to suggest that there are other similar childhood obesity programs in the community (there are not), but to emphasize that parents were part of a special and unique group (i.e., C.H.A.M.P. Families) with a shared focus, and that this likely contributed to group cohesion (through the group dynamics strategy ‘distinctiveness’).
Discussion
Point 17: Line 644 suggest to replace via with through
Response 17: Thank you. Amended as suggested (Line 886).
Conclusion:
Point 18: Seems to relate to section 2 of the protocol paper only, suggest to amend.
Response 18: Thank you for this observation. We have revised to ensure that the Conclusion relates to both the description of the intervention as well as the theoretical framework (see Page 19, Lines 900-901).
References:
Point 19: Please amend references with long and institutional URLs – e.g. 125, 122, 71. Instead use DOI URLs or even cache if appropriate
Response 19: Thank you. As per the reviewer’s suggestion, we have revised the following references: 2, 12, 33, 51, 52, 54, 58, 59, 73, 86, 89, 104

Reviewer 2 Report
This manuscript sets out the implementation and parts of the study protocol, including extensive justification for the theoretical basis of the intervention, for a single-arm, single-centre feasibility study to investigate the CHAMP intervention for reducing childhood (age 6–14) overweight and obesity (age- and sex-standardised BMI z-score>1.04 for entry into the study). The intervention involved a comprehensive mixture of components including nutrition, physical activity, sleep, and mental health. A mixture of group- and home-based components are described, with a strong parent-focus throughout.
The well-written introduction is sufficiently comprehensive and motivates the present study. I wondered if readers would fully appreciate that when you compare parent-only versus parent and child or child-only interventions (e.g. Lines 75–79 and 85–87), the issue is, I think, not that child-involvement is negative (although it certainly could be in some cases) but rather seems related to how limited resources are spent. I absolutely agree with the parent-focus of your study, I just wondered if Line 87 might be misinterpreted as suggesting that child-involvement is negative rather than that the evidence is that this provides less returns than similar or even lower investments in parent-focused interventions. I think the following text in your introduction should make the actual point clear but you could add “similarly resourced” (or “similar intensity”) before “program” on Line 87 if you agree that this would make it clearer.
I wonder also, and as a disclosure the trials I’ve been involved with in this area have mostly looked at interventions starting pre-birth, whether the current interest in the “first 1000 days” is worth explicitly mentioning in the introduction. If you accept that approach, studies such as yours are crucial in helping those children who become overweight or obese, which is especially important given the limited success of such prevention approaches, but they become a second wave after the first wave of prevention. I don’t think you absolutely need this small addition, but it would help place your study in context from my, admittedly biased, perspective.
I disagree with the argument in section 2.1.4 (Sample size). The sample size for feasibility studies needs to be sufficient to achieve one or more of a) assess aspects of the protocol (including recruitment, delivery, retention, assessment, and statistical analysis) and allow refinements as needed prior to larger studies (which may be phrased around achieving saturation of any issues), b) detect adverse events (e.g. the “rule of three”), c) provide information about parameters needed to determine the sample size for a larger study, and d) to provide an initial indication of effectiveness or futility even though this is very rarely definitive and only with an appropriate control/comparison group. A formal sample size statement around detecting the smallest practical difference would indeed be inappropriate, but the feasibility study still needs to achieve at least one of the goals mentioned above to be of value.
In order for the manuscript to serve its initially apparent purpose as a study protocol, I would like to see more information on the anthropometry, including models and calibration of equipment, training of anthropometrists, use of multiple measurements (presumably 2 with a third where there was a discrepancy?), and any inter- or intra-rater reliability checks. Similarly, the protocol for the Acticals needs explaining (site worn, instructions, and data cleaning rules including non-use). Finally, statistical and qualitative analyses should be described in brief. With regards to the former, while I appreciate that analyses here must differ from a larger RCT, the sample size would still permit linear mixed models to investigate changes over time and predictors of this. If, however, the term “protocol” is being used strictly in the sense of “implementation protocol”, this should be made clearer (e.g. Lines 2, 14, 19, 136, 140). Other references to “protocol” seem to be clearly limited to the implementation. However, if the focus is on an implementation protocol, there seems to be more detail than necessary around the study protocol.
I appreciate that this may be planned for a subsequent manuscript, but some “lessons learned” or “limitations” (from the outset or with hindsight) would be useful additions in the Discussion.
My main concern is that with the intervention delivered in 2017 and the final “booster” session and data collection in June 2018, as a protocol in general, this loses one of its main purposes of pre-specifying details of study hypothesis, data processing, and analyses prior to the completion of data collection (again assuming that there is at least an element of study protocol here). I appreciate that the theoretical aspects are well worth describing and the protocol will still assist with publishing articles from the study, but I do have a degree of discomfort with such late study protocol publications if this is the intention here.
Specific comments:
Lines 23–24: “a practical example” (or “practical examples”)
Line 39: Closing parenthesis is missing (opening on Line 38).
Line 42: Closing parenthesis is missing (opening on Line 41).
Line 203: I appreciate that “PowerPoint” has become a genericised trademark for slide presentations, but I’d suggest that “slide presentation” would be more accurate (outside of the unlikely situation where a PDF-based presentation, for example, would not be acceptable).
Line 209: “MI” has not previously been defined as an abbreviation.
Table 1: Semi-colons in the final column are mostly unnecessary (these would only be needed if the list items themselves contained sub-lists or clauses using commas). The one exception might be the 4th session but I’m unclear why sedentary activity and screen time needed to be listed twice here. The same point applies elsewhere (e.g. Lines 272–280 where the commas are all parenthetical).
Line 272: “in THE intervention” (or “an”)
Table 2: Was it actually gender (social construct) that was elicited for parents and children and not biological sex? This seems particularly challenging for the latter given age- and sex-standardised z-scores were used.
Table 2: An instance of “HRQOL” should be “HRQoL” for consistency with the text and the other mention in the table.
Lines 362–363: While either is acceptable, you use “BMI-z” (e.g. Lines 56, 91, 124, and 283, and Table 2; although I’d prefer “BMI z-score”) elsewhere so I’d change “BMI standard deviation scores” to this. It would be useful to explicitly note the source of the z-scores at some stage.
Table 3: I think “Collection of reflective statements that drawn together” should include “draw together” (deleting the “n”) or the following “and” deleted.
Line 455: I think there is a comma missing before “ultimately”.
Lines 489–490: I don’t think you need to provide S.M.A.R.T. in full again here (already done on Lines 239–240).
Line 574: Presumably you mean “sitting in the same seats across sessions”?
Author Response
Response to Reviewer 2 Comments
This manuscript sets out the implementation and parts of the study protocol, including extensive justification for the theoretical basis of the intervention, for a single-arm, single-centre feasibility study to investigate the CHAMP intervention for reducing childhood (age 6–14) overweight and obesity (age- and sex-standardised BMI z-score>1.04 for entry into the study). The intervention involved a comprehensive mixture of components including nutrition, physical activity, sleep, and mental health. A mixture of group- and home-based components are described, with a strong parent-focus throughout.
The well-written introduction is sufficiently comprehensive and motivates the present study. I wondered if readers would fully appreciate that when you compare parent-only versus parent and child or child-only interventions (e.g. Lines 75–79 and 85–87), the issue is, I think, not that child-involvement is negative (although it certainly could be in some cases) but rather seems related to how limited resources are spent. I absolutely agree with the parent-focus of your study, I just wondered if Line 87 might be misinterpreted as suggesting that child-involvement is negative rather than that the evidence is that this provides less returns than similar or even lower investments in parent-focused interventions. I think the following text in your introduction should make the actual point clear but you could add “similarly resourced” (or “similar intensity”) before “program” on Line 87 if you agree that this would make it clearer.
Response: Thank you, this is a great (and accurate) observation. As per the reviewer’s suggestion, we have revised to include “similar intensity” before the word program to avoid any misinterpretations (please see Page 2, Lines 93-94).
I wonder also, and as a disclosure the trials I’ve been involved with in this area have mostly looked at interventions starting pre-birth, whether the current interest in the “first 1000 days” is worth explicitly mentioning in the introduction. If you accept that approach, studies such as yours are crucial in helping those children who become overweight or obese, which is especially important given the limited success of such prevention approaches, but they become a second wave after the first wave of prevention. I don’t think you absolutely need this small addition, but it would help place your study in context from my, admittedly biased, perspective.
Response: This is a very good point and certainly an interesting area of research. We also acknowledge the significance of the first 1000 days. However, because we were asked by a reviewer to reduce the length of the intervention (and also to include additional content based other recommendations), we simply did not have room to include such information. No action taken.
I disagree with the argument in section 2.1.4 (Sample size). The sample size for feasibility studies needs to be sufficient to achieve one or more of a) assess aspects of the protocol (including recruitment, delivery, retention, assessment, and statistical analysis) and allow refinements as needed prior to larger studies (which may be phrased around achieving saturation of any issues), b) detect adverse events (e.g. the “rule of three”), c) provide information about parameters needed to determine the sample size for a larger study, and d) to provide an initial indication of effectiveness or futility even though this is very rarely definitive and only with an appropriate control/comparison group. A formal sample size statement around detecting the smallest practical difference would indeed be inappropriate, but the feasibility study still needs to achieve at least one of the goals mentioned above to be of value.
Response: Again, thank you for this comment. The primary objective of this study was to evaluate the feasibility of C.H.A.M.P. Families using the RE-AIM Framework (Glasgow, Vogt, & Boles, 1999), which takes into consideration several aspects of the first goal described above (i.e., reach [recruitment and retention], preliminary effectiveness, implementation [delivery]) as well as others (i.e., adoption, maintenance). Prior to the start of the intervention, we aimed (ideally) to recruit 30 parent-child dyads (or triads, if both parents/caregivers were interested in participating) to participate in the program. Following four months of heavy recruitment using several different strategies (May-September, 2017), we had recruited 11 eligible families (6 dyads and 5 triads; n = 11 children and 16 parents/caregivers), and thus, we proceeded with the program as planned. With regard to the specific considerations for sample size, we will evaluate the “Reach” of the intervention in several ways: (1) comparing the demographics of our participants to the census demographics of our city, (2) investigating reasons for participating or declining to participate, (3) determining most effective recruitment methods, and (4) calculating participation rate. A full summary of the feasibility outcomes and measures (including timeline) can be found in Table 2 (Pages 9-10).
In order for the manuscript to serve its initially apparent purpose as a study protocol, I would like to see more information on the anthropometry, including models and calibration of equipment, training of anthropometrists, use of multiple measurements (presumably 2 with a third where there was a discrepancy?), and any inter- or intra-rater reliability checks. Similarly, the protocol for the Acticals needs explaining (site worn, instructions, and data cleaning rules including non-use). Finally, statistical and qualitative analyses should be described in brief. With regards to the former, while I appreciate that analyses here must differ from a larger RCT, the sample size would still permit linear mixed models to investigate changes over time and predictors of this. If, however, the term “protocol” is being used strictly in the sense of “implementation protocol”, this should be made clearer (e.g. Lines 2, 14, 19, 136, 140). Other references to “protocol” seem to be clearly limited to the implementation. However, if the focus is on an implementation protocol, there seems to be more detail than necessary around the study protocol.
Response: Thank you. Upon consideration of the reviewer’s suggestions and comments, and further reflection/refining of our original purpose, we have now clarified throughout the manuscript (e.g., Page 3, Lines 157-160; Page 4, Lines 177-180, Line 209; Page 18, Lines 855-856; Page 19, Lines 880-884)—including a revised title—that our intent was to provide a detailed description of the C.H.A.M.P. Families intervention as well as its theoretical underpinnings, rather than to provide an overview of the study protocol per se. We feel that these revisions have strengthened the manuscript, and that the information provided within the paper will fill some important gaps in the literature related to the (under)reporting of both intervention components (Hermann et al., 20107; Loveman et al., 2015) and underlying theoretical frameworks (Alulis & Grabowski, 2017). We have also selected the Template for Intervention Description and Replication (TIDieR) checklist and guide (Hoffman et al., 2014) as the most appropriate guide for this work. The application of this checklist also compliments the secondary purpose of the manuscript, which is to describe the theoretical framework underpinning the intervention and to provide practical examples of how theoretical constructs and evidence-based strategies can be applied in future parent-focused paediatric obesity interventions.
Specifically, as per the reviewer’s comments and suggestions, we have revised several areas of the manuscript to clarify that it is not a study protocol (Page 3, Lines 157-160; Page 4, Lines 177-180) and have removed the study protocol-related information (i.e., participant recruitment [Page 4], sample size [Page 4], Primary and Secondary Outcomes in Table 2 [Pages 9-10]) deemed unnecessary for the purposes of this manuscript. Given the broad scope of this mixed-methods feasibility study, we have decided that we will present specific methodological and data analysis information within the manuscripts that will focus on evaluating the effectiveness of those outcomes. This decision will also ensure that the current manuscript is succinct and focused on describing the intervention and theoretical framework, as opposed to the study protocol. The complete TIDieR checklist is also now available as a Supplementary File.
I appreciate that this may be planned for a subsequent manuscript, but some “lessons learned” or “limitations” (from the outset or with hindsight) would be useful additions in the Discussion.
Response: We do plan to provide such information in subsequent manuscripts, however given the reviewer’s suggestion, we have now included a brief section on lessons learned (see Page 19, Lines 870-876).
My main concern is that with the intervention delivered in 2017 and the final “booster” session and data collection in June 2018, as a protocol in general, this loses one of its main purposes of pre-specifying details of study hypothesis, data processing, and analyses prior to the completion of data collection (again assuming that there is at least an element of study protocol here). I appreciate that the theoretical aspects are well worth describing and the protocol will still assist with publishing articles from the study, but I do have a degree of discomfort with such late study protocol publications if this is the intention here.
Response: This is a valid concern that was considered very carefully. As stated above, we have now revised the purpose statement (Page 3, Lines 157-160) and descriptions of the study throughout the manuscript to highlight that rather than a protocol, we are providing a detailed description of the intervention and its underlying theoretical foundation. In addition, and also noted above, the application of the TIDieR checklist (Hoffman et al., 2014) was deemed more appropriate for this purpose than the CONSORT extension for Pilot and Feasibility Studies and these sections have also been revised (Page 4, Lines 177-181). As per the TIDieR guide specific information pertaining to the intervention providers (Lines 448-466), location (line 251), dose (Line 252-259, 428), and adherence (Lines 271-273, Lines 304-307) was added. Information that related to the study protocol and was not necessary to describe the intervention was removed (i.e., participant recruitment [Page 4], sample size [Page 4], Primary and Secondary Outcomes in Table 2 [Pages 9-10])
Specific comments:
Lines 23–24: “a practical example” (or “practical examples”)
Response: We have now revised to “practical examples” (Line 23).
Line 39: Closing parenthesis is missing (opening on Line 38).
Response: Thank you. In an effort to reduce the length of the manuscript, we have now removed this line from the Introduction so this is no longer an issue.
Line 42: Closing parenthesis is missing (opening on Line 41).
Response: Thank you. In an effort to reduce the length of the manuscript, we have now removed this line from the Introduction so this is no longer an issue.
Line 203: I appreciate that “PowerPoint” has become a genericised trademark for slide presentations, but I’d suggest that “slide presentation” would be more accurate (outside of the unlikely situation where a PDF-based presentation, for example, would not be acceptable).
Response: Good point. We have now revised “PowerPoint” to “slide presentation” (Line 276).
Line 209: “MI” has not previously been defined as an abbreviation.
Response: We have now revised to include “motivational interviewing” prior to first use of abbreviation (Line 453).
Table 1: Semi-colons in the final column are mostly unnecessary (these would only be needed if the list items themselves contained sub-lists or clauses using commas). The one exception might be the 4th session but I’m unclear why sedentary activity and screen time needed to be listed twice here. The same point applies elsewhere (e.g. Lines 272–280 where the commas are all parenthetical).
Response: Thank you. Sedentary activity and screen time were repeated in Table 1 unintentionally and have been removed. Additionally, we have replaced semi-colons with commas as per the suggestion in Table 1 (Pages 6-7) Lines 204-208, 201-215, 306-311, 491-499, 580-583
Line 272: “in THE intervention” (or “an”)
Response: In this sentence we are describing the motivational spirit and techniques that were applied by the research team in this specific intervention (i.e., “the”). No action taken.
Table 2: Was it actually gender (social construct) that was elicited for parents and children and not biological sex? This seems particularly challenging for the latter given age- and sex-standardised z-scores were used.
Response: Thank you for raising this point. The word “gender” was included unintentionally. We collected biological sex information for both parents and children (parent-reported). Table 2 (Pages 9-10) has been edited accordingly.
Table 2: An instance of “HRQOL” should be “HRQoL” for consistency with the text and the other mention in the table.
Response: Thank you. We have now removed that section of Table 2 so this is no longer an issue.
Lines 362–363: While either is acceptable, you use “BMI-z” (e.g. Lines 56, 91, 124, and 283, and Table 2; although I’d prefer “BMI z-score”) elsewhere so I’d change “BMI standard deviation scores” to this. It would be useful to explicitly note the source of the z-scores at some stage.
Response: We have revised “BMI standard deviation scores” to “BMI-z” throughout to keep consistent with the rest of the manuscript (Page 11, Line 609). The source for BMI-z scores was the Centers for Disease Control and Prevention 2000 Growth Charts (https://www.cdc.gov/growthcharts/percentile_data_files.htm). However, since we have removed the specific information related to the data analysis of primary and secondary outcome, this information will be reported in a subsequent manuscript.
Table 3: I think “Collection of reflective statements that drawn together” should include “draw together” (deleting the “n”) or the following “and” deleted.
Response: Thank you. As per your suggestion, we have removed the “and” and revised to “statements that drawn together suggest links…” (Table 3, Pages 12-13).
Line 455: I think there is a comma missing before “ultimately”.
Response: This sentence has now been revised (Line 709).
Lines 489–490: I don’t think you need to provide S.M.A.R.T. in full again here (already done on Lines 239–240).
Response: Thank you. We agree and have removed the repetition here (Line 743).
Line 574: Presumably you mean “sitting in the same seats across sessions”?
Response: Yes, we have now provided clarification here (Lines 829-830).
